# Power-Frequency Electric Field Sensing Utilizing a Twin-FBG Fabry–Perot Interferometer and Polyimide Tubing with Space Charge as Field Sensing Element [note 1]

**DOI:** 10.3390/s19061456

**Published:** 2019-03-25

**Authors:** Lutang Wang, Nian Fang

**Affiliations:** Key Laboratory of Specialty Fiber Optics and Optical Access Networks, School of Communication and Information Engineering, Shanghai University, Shanghai 200444, China; nfang@shu.edu.cn

**Keywords:** electric field, fiber optic sensor, polyimide tubing, dielectric charging, space charge

## Abstract

A novel fiber-optic sensor based on the alternating electric field force actions on polyimide tubing with space charge for power-frequency electric field sensing is presented. In structure, the sensor consists of a lightweight fiber cantilever beam covered with a length of electrically charged polyimide tubing as the field sensing element. A twin-FBG based Fabry–Perot interferometer is embedded in this fiber beam to detect the beam vibrations excited by the force of power-frequency electric field to be sensed. Space charge in polyimide tubing is formed through a dielectric charging process. The basic concept, structure, fabrication and operation principle of the sensor are introduced with detailed theoretical analyses. The comprehensive experiments with two sensor prototypes are carried out, in which a sensor exhibits a high sensitivity of 173.65 μV/(V/m) with a minimal detectable field strength of 0.162 V/m, and another has a durability of continuous operation for over a year.

## 1. Introduction

In the power industry, the healthy conditions of a high-voltage (HV) power equipment, referring to voltages, currents, temperatures and electric field (*E*-field) strengths at power frequency (50 or 60 Hz) as well as vibrations and partial discharges, need to be monitored in real time [1,2,3,4,5,6]. So far, several different types of *E*-field and voltage sensors based on electrical or optical detection schemes have been proposed and developed [7,8,9,10]. Optical or fiber-optic types of sensors usually can offer many advantages over conventional electrical types, such as the immunity to the electromagnetic interference (EMI) and a capability to achieve the remote sensing [11].

To achieve field strength sensing, most of optical *E*-field sensors utilize the electro-optic (EO) effects such as Pockels and Kerr effects existing in some crystals, and electrostrictive effects existing in some polymers and piezoelectric materials [12,13,14,15,16]. These physical effects modulate the birefringence of EO materials or the effective refractive index of optical fiber, which can be detected with the most common methods such as using fiber-optic interferometers [14,15] or polarimeters. In structure, most of previously developed EO sensors require two polarization maintaining fibers (PMF) connected to the device as input/output outlets of the probe light, which often causes the sensor in difficulty in installations as well as in reducing whole size. Recently, with developments of lithographic and micro/nano-machining technologies, a variety of state-of-the-art, light planar waveguide or glass fiber based EO sensors have been developed [17,18,19,20]. Among these new types of EO sensors, some already adopted the novel structures allowing to use a single fiber for probe light transmissions [18,19], which considerably reduced the whole size of the sensor. However, a metallic dipole antenna (electrodes) to generate induction voltage to drive the device to work still was needed. The electrodes usually were deposited directly on the device, which made these small-size (in millimeter scale) EO sensors must work in higher frequency ranges, e.g., in the RF range. It is obviously not suitable for *E*-field sensing in the power-frequency range.

Utilizing the induced electric field force to move or deform an electrically charged object has been considered as an alternative sensing method for the detection of static or low-frequency *E*-field strengths. As a typical implementation of this method, in recent years, a variety of MEMS technology based *E*-field sensors have been developed. A MEMS type of *E*-field sensor with a sensitivity of 0.3 V/m (minimum detectable field strength) at 97 Hz had been reported [21]. Other MEMS structure *E*-field sensors, e.g., using the Fabry–Perot (FP) interferometer for DC high voltage measurements, also had been proposed [22]. In structure, the MEMS *E*-field sensor usually needs a parallel, electrically grounded metallic electrode as a zero-potential reference. The air gap between the reference electrode and membrane is so small (only in micrometer scale) that in HV environments, the electric discharges between electrodes often occur, which inevitably degrades the reliability of the sensor [23]. In addition, in some applications, where the sensors have to be installed at a higher place far above the ground, the use of zero-potential reference may cause a lot of inconvenience.

In our previous experimental work, we had investigated the feasibility of a power-frequency fiber-optic *E*-field sensor by utilizing alternating electric field forces to excite a lightweight fiber cantilever beam to vibrate at power frequency. In our approach, instead of using a metal membrane for field sensing, a dielectric polyimide (PI) tubing with space charge covering on the fiber cantilever beam was adopted as the field sensing element. Primary experimental results for demonstrating this approach had been reported in [24]. In this article, based on the previous achievements, the detailed theoretical analyses in respect of the sensor’s structure, dielectric charging and operation principle will be presented in Section 2. Sensor fabrications and the detection system are introduced in Section 3. More comprehensive experimental results related to charging assessments and characterizations of the sensor will be demonstrated in Section 4. The results referred to actual field sensing and applications in other measurement fields will be presented in Section 5. Finally, conclusions are given in Section 6.

## 2. Sensor Structure, Dielectric Charging and Operation Principle

### 2.1. Sensor Structure

Figure 1a shows a schematic on our *E*-field sensor structure. It employs a piece of PI resin coated single-mode optical fiber covered (jacked) with a length of PI tubing with space charge inside (Figure 1b), clamped at a position with a distance from its free end, to constitute a lightweight, composite fiber cantilever beam as the field sensing element. Two identical fiber Bragg gratings (twin-FBG) with the low reflectivity and an interval $L_{FP}$ are imprint on the optical fiber close to the end of the beam to form a low-fineness, in-line FP interferometer (FP sensor). In the presence of the AC *E*-field, the pre-electrically charged PI tubing can get the lateral *E*-field force to excite the cantilever beam to vibrate at AC frequency, as shown in Figure 1c, with an amplitude proportional to the field strength, which is detected by means of FP sensor. In this structure, PI tubing as an *E*-field sensitive medium plays a key role.

As a group of dielectric polymers, PI materials with a dielectric constant of ∼3.4 in the 100-Hz range exhibiting outstanding engineering properties, especially the thermal stability, dielectric and mechanical strengths and chemical resistance, have been widely employed in different engineering fields for electrical insulation, optical fiber coating, electromechanical transduction [25], etc. In our research, however, we exploit another potential application of PI materials as a sort of field sensitive medium for *E*-field sensing.

According to the band theory, polymers possess a capability to store charges trapped in a dielectric charging process which often is referred to an electrode charging or a carrier-injection charging [26,27]. During the charging process, under *E*-field and temperature influences, external carriers (electrons or holes) from electrodes or injection can transport from the surface of polymer sample into its volume and be trapped by the impurities and defects existing inside materials and become the volume charges or called space charge [26,28]. Space charge prefers to accumulate at the interfaces of two adjacent layers due to permittivity/conductivity difference [27,29], and spatially distributes near the sample surface with an average depth in several ten microns, which mainly depends on the injection energy, charging time, temperature and trap density [27,28]. Under normal environmental conditions where ambient humidity is relatively low and temperature is not higher than 120 °C (a temperature point to start the charge loss), space charge can stably stay in the polymer for a long time (even up to $10^{12}$ s) without obvious decreases in the density [27]. On the other hand, PI materials also are one of polar dielectric polymers [30] containing numerous electric dipoles with random orientations without any movements [31]. So, PI materials normally present non-electrical property even in the presence of external electric fields. However, as the temperature is high enough, these inherent electric dipoles will move and reorient to the direction of external electric field [31]. When the temperature is reduced rapidly and the external electric field is removed, the electric dipoles will stop moving and statistically align in one direction, which makes PI materials appear the electrical polarization property [25,32]. Basically, the electrical polarization in polymers is generally produced by either space charge or dipole orientation. In brief, after dielectric charging, PI materials appear obvious electrical behaviors with both features of space charge and electric dipole, which can be utilized as field sensitive mediums.

PI tubing is commercially available and a product series of MicroLumen Inc., classified by the inner diameter with a specific production code. In our experiments, usually adopted codes were #068 and #085 with outer diameters of 270 μm and 320 μm as well as inner diameters of 180 μm and 220 μm, respectively. Which code should be selected entirely depends on the actual diameter of the fiber grating sensor to be used. Usually, for a fiber grating sensor with PI resin coating, its diameter varies from 150 μm to 200 μm. It should be noted that PI tubing constructed by MicroLumen Inc. possesses a few of laminated layers formed through a well-known *dipping* processing, which yields numerous interfacial regions inside the tubing, which are particularly favorable to the space charge formation and storage. In addition, PI tubing as an elastomer possesses outstanding mechanical properties (the flexibility and stiffness) and damping characteristics which can effectively protect the glass fiber as the beam vibrating in a resonant mode with considerably large amplitude.

### 2.2. Dielectric Charging

For implementing the dielectric charging of PI tubing, we designed a jig as shown in Figure 2a, which consists of two copper blocks (cover and substrate) with V-grooves to hold PI tubing as an electrode (electrode *I*) and a wire inserted into PI tubing as another electrode (electrode *II*). A TEC (thermoelectric cooler) device attached to the substrate controls the temperature of the jig during charging. A HV DC voltage is imposed on two electrodes to form a high-strength external electric field for charging. Generally, after charging, an evaluation for charging effects is necessary, which is generally based on a measurement of the charge profile in terms of the density and temporal evolution of trapped charges including surface charge and space charge. The measuring methods usually employed in investigating of the insulation aging and failure mechanism of polymer materials [28,33,34], however, are complicated and unsuitable for our purpose. Hence, as an alternative, we designed a simple measuring procedure as schematically shown in Figure 2b for statistically assessing charging effects. In principle, this procedure is to measure an alternating *E*-field generated by moving charges in space, which makes of use of a mechanical shaker to drive the charged PI tubing to vibrate at a specific frequency *f* and an *E*-field meter to measure the surrounding *E*-field strength. The *E*-field near PI tubing at *p* point in space is a superposed field, generated by all electric charges in PI tubing distributing at different positions and in different depths. According to Coulomb’s law, the alternating *E*-field created by moving charges in space through PI tubing vibrations can be approximately expressed as
(1)$$E_{ac}\left(t\right)=\frac{1}{4\pi \varepsilon_{0}}\left(\sum_{i}\frac{Q_{i}\Delta r_{i}}{r_{i}^{3}}\right)\sin\omega t$$
where ω=2πf is the circular frequency of PI tubing vibrating, $\varepsilon_{0}$ is the permittivity of free space, $Q_{i}$ is a sum of all positive and negative charges (holes and electrons) at region $s_{i}$, $r_{i}$ is a static-state distance between $s_{i}$ and *p*, and $\Delta r_{i}$ is the maximum variation (vibrating amplitude) of $r_{i}$. From Equation (Equation 1), it is clear that when the mechanical vibration of PI tubing is steady, during a short measurement period, the field strength $|E_{ac}\left(t\right)|$ will be a function of the number of total charges staying in PI tubing, which can be used to evaluate the charge density in PI tubing. When measurement time is long enough, the decay of $|E_{ac}\left(t\right)|$, that is the temporal evolution of trapped charges, which relates to a discharging/detrapping process of electric charges [27], also can be observed. It should be noted that although this procedure is simple in operation, the detailed knowledge about the mobility, trapping depth and distribution states of space charge in PI tubing is still not clear, which may be obtained by means of other sophisticated methodologies, e.g., the pulsed electro-acoustic (PEA) or the laser induced pressure propagation (LIPP) techniques [33,34,35,36]. However, these are out of our research scope. The relevant experimental results will be presented in Section 3.

### 2.3. Operation Principles

For designing an all-fiber *E*-field sensor capable of working in the power frequency range, the vibration sensor based on a cantilever beam, or called, cantilever system may be a good reference. Silica glass based optical fiber itself is an excellent elastomer, so that it can be utilized to constitute a lightweight cantilever beam. After being jacked with the charged PI tubing, a composite cantilever beam with *E*-field sensitive property can be formed. A schematic on this concept is shown in Figure 3, in which *m* and *L* represent the mass and length of this composite cantilever beam, respectively. As mentioned above, in a high-strength, power-frequency *E*-field, the AC *E*-field force induced by space charge and electric dipoles in PI tubing can excite the cantilever beam to vibrate at power frequency. Assuming that space charge and electric dipoles uniformly distribute along the length of PI tubing, in the presence of an AC *E*-field with power frequency $f_{e}$, the induced AC *E*-field force can be expressed as
(2)$$F\left(t\right)=\Sigma QE_{0}\left(t\right)+\mu \nabla E_{0}\left(t\right)=F_{1}\left(t\right)+F_{2}\left(t\right)$$
where ΣQ denotes the assembled space charge, μ is the assembled electric dipoles, $E_{0}\left(t\right)$ is the local *E*-field and $\nabla E_{0}\left(t\right)$ is the local *E*-field gradient. F(t) consists of two components: $F_{1}\left(t\right)$ directly from the space charge contributions, relating to $E_{0}\left(t\right)$, and $F_{2}\left(t\right)$ from the contributions of electric dipoles, relating to $\nabla E_{0}\left(t\right)$. In a uniform field or when the electric dipoles are negligible, having $F_{2}\left(t\right)=0$. Compared to the space charge effect, generally, the electric dipole effect in PI tubing is relatively small, so that, for simplicity, in following analyses, we assume $F_{2}\left(t\right)=0$, $F\left(t\right)=F_{1}\left(t\right)$. F(t) is parallel in space to $E_{0}\left(t\right)$, while its direction is determined by the polarity of ΣQ.

For getting the detailed analysis results on the motion of a cantilever beam subjected to an AC *E*-field, we assume that the displacement of neutral axis of the beam is small, and neglect the gravity effect. Consider a coordinate in Figure 3, the static displacement of the beam at any location *x*, parallel to the *y*-axis, generated by the uniformly distributed lateral static *E*-field force, is given by [37]
(3)y(x)=Af(x)
with
(4)$$A=\frac{F_{l}L^{3}}{8(EI)},\;f\left(x\right)=\frac{1}{3}{\left(\frac{x}{L}\right)}^{4}-\frac{4}{3}{\left(\frac{x}{L}\right)}^{3}+2{\left(\frac{x}{L}\right)}^{2}$$
where *A* is the maximum displacement at the free end of the beam and f(x) is a distribution function of the beam displacement along the *x*-axis, having f(0)=0 and f(L)=1. $F_{l}$ is the magnitude of lateral *E*-field force $F_{l}\left(t\right)$, having $F_{l}=|F_{l}\left(t\right)|=|F\left(t\right)|\sin\theta$, here θ is an angle between F(t) and the beam. (EI) is the flexural rigidity of a composite beam, having $\left(EI\right)=E_{a}I_{a}+E_{b}I_{b}$ [38], where $E_{a}I_{a}$ and $E_{b}I_{b}$ are the flexural rigidity, furthermore $E_{a}$ and $E_{b}$ being the Young’s modulus, while $I_{a}$ and $I_{b}$ being the 2nd moment of area, of glass fiber and PI tubing, respectively (see Figure 3).

An *E*-field sensor based on the cantilever beam structure can be modeled as a forced viscous-damping vibration system with one degree of freedom [39]. Assuming that the beam is subjected to an AC *E*-field $E_{0}\left(t\right)$ and vibrates in the xy coordinate plane as shown in Figure 3 at $\omega_{e}=2\pi f_{e}$ with a phase ϕ lag to $E_{0}\left(t\right)$, when the vibration amplitude is relatively low over a large range of exciting *E*-field force, both the beam dynamic response and the induced fiber axial strain will be the linear functions of the lateral AC *E*-field force. The motion equation of a cantilever system used to describe the dynamic equilibrium of overall forces acting on the beam at any location *x* and any time *t* is given by [39]
(5)$$\frac{\partial^{2}y\left(x,t\right)}{\partial t^{2}}+2\zeta \frac{\partial y(x,t)}{\partial t}+\omega_{0}^{2}y\left(x,t\right)=\frac{f_{l}x}{m}\cos\omega_{e}t$$
where ζ is the coefficient of viscous damping and $f_{l}=F_{l}/L$ is the intensity of lateral *E*-field force. $\omega_{e}$ and $\omega_{0}$ are the circular frequency of the AC *E*-field and the natural circular frequency of the cantilever system, respectively. Assuming $y\left(x,t\right)=u\left(x\right)\cos\left(\omega_{e}t-\phi \right)$ being a solution to Equation (Equation 5), we have
(6)$$\begin{array}{cc} \frac{\partial y(x,t)}{\partial t} & =-\omega_{e}u\left(x\right)\sin\left(\omega_{e}t-\phi \right)=\omega_{e}u\left(x\right)\cos\left(\omega_{e}t-\phi +\frac{\pi }{2}\right)\\ \frac{\partial^{2}y\left(x,t\right)}{\partial t^{2}} & =-\omega_{e}^{2}u\left(x\right)\cos\left(\omega_{e}t-\phi \right)=\omega_{e}^{2}u\left(x\right)\cos\left(\omega_{e}t-\phi +\pi \right) \end{array}$$
by substituting these into Equation (Equation 5) and taking a force vector addition operation [39], we obtain
(7)$$u\left(x\right)=\frac{f_{l}x}{m\omega_{0}^{2}}{\left[{\left(1-\eta^{2}\right)}^{2}+4\zeta^{2}\eta^{2}\right]}^{-1/2}$$
and
(8)$$\tan\phi =\frac{2\zeta \eta }{1-\eta^{2}}$$
where $\eta =\omega_{e}/\omega_{0}$ is a dimensionless frequency parameter. Since $\omega_{0}^{2}=K/m$, where *K* is the beam stiffness and $K=F_{l}/y\left(L\right)=8\left(EI\right)/L^{3}$, we have $m\omega_{0}^{2}=8\left(EI\right)/L^{3}$. Hence, Equation (Equation 7) can be rewritten as
(9)$$u\left(x\right)=\frac{f_{l}xL^{3}}{8(EI)}{\left[{\left(1-\eta^{2}\right)}^{2}+4\zeta^{2}\eta^{2}\right]}^{-1/2}$$

According to Equation (Equation 9), the steady-state vibration amplitude of a cantilever system is defined as the static displacement of the beam at the free end (x=L) when η=0, having $u\left(L\right)=F_{l}L^{3}/\left[8\left(EI\right)\right]\;=\;A$, so that we obtain u(x)=Af(x)=y(x) when η=0. Hence, the general dynamic response of the cantilever system can be expressed as
(10)$$y\left(x,t\right)=y\left(x\right){\left[{\left(1-\eta^{2}\right)}^{2}+4\zeta^{2}\eta^{2}\right]}^{-1/2}\cos\left(\omega_{e}t-\phi \right)$$

According to Equation (Equation 8), when η=1, having ϕ=π/2, the cantilever system works in a resonance mode. So, Equation (Equation 10) becomes
(11)$$y\left(x,t\right)=y\left(x\right)Q\sin\omega_{e}t$$
where Q=1/(2ζ) [40] is the *Q*-factor or the amplification coefficient of a resonant system. It is obvious that if Q>1, the static displacement of the beam, y(x), can be dynamically amplified to y(x)Q. In our sensor design, in order to enhance the sensitivity, we usually select a beam length so as to let $\omega_{0}=\omega_{e}$. This particular beam length is determined by
(12)$$L={\left[\frac{8(EL)}{m\omega_{e}^{2}}\right]}^{1/3}$$

By changing the pressure at the clamp point, the damping parameter ζ can be adjusted. Generally, the larger the pressure is, the higher is the value of *Q*. In this way, the sensor sensitivity is controllable. In order to prevent the beam damage as the cantilever system works in a resonant mode, the maximum vibrating amplitude of the beam should be limited to a level determined by a dimensionless vibration intensity parameter *S*, defined as [40]
(13)$$S=10\log\frac{Z}{Z_{0}}$$
where *Z* is the beam vibration intensity defined with $Z=2\pi |d^{2}y\left(L,t\right)/dt^{2}{|}^{2}/\omega_{e}=2\pi Q^{2}{\left|y\left(L\right)\right|}^{2}\omega_{e}^{3}$, and $Z_{0}$ is a reference value usually taken to be 10 ${\mathrm{mm}}^{2}$/s${}^{3}$. According to the calculated results in [40], when S<17.5, the vibrations of a beam can be considered to be safety.

Fiber axial strain induced by beam vibrating is directly related to the curvature of neutral axis of the beam [39,41], which in a resonant state, with Equation (Equation 11), can be approximately expressed as
(14)$$\varepsilon \left(x,t\right)=r_{a}\frac{\partial^{2}y\left(x\right)}{\partial x^{2}}Q\sin\omega_{e}t=\frac{r_{a}F_{l}Q{\left(L-x\right)}^{2}}{2L(EI)}\sin\omega_{e}t$$
where $r_{a}$ is the radius of the fiber. Clearly, ε(x,t) is a periodic function of the time with an oscillating frequency $\omega_{e}$ and a location-dependent magnitude which achieves the maximum at x=0. A location average axial strain over the whole FP sensor length *D* at a given location $x_{0}$ ($x_{0}\geq D/2$) is obtained
(15)$$\varepsilon \left(t\right)=\frac{1}{D}\int_{x_{0}-D/2}^{x_{0}+D/2}\varepsilon \left(x,t\right)dx=\frac{3r_{a}F_{l}Q}{8L(EI)}\left[4{\left(L-x_{0}\right)}^{2}+D^{2}\right]\sin\omega_{e}t$$

As beam vibrating, ε(t) modulates the Bragg wavelength of FP sensor through periodically elongating or compressing the grating and changing the effective refractive index of the fiber [42]. The amount of wavelength variation is given by
(16)$$\Delta \lambda \left(t\right)=\lambda_{B0}\left(1-P_{e}\right)\varepsilon \left(t\right)$$
where $P_{e}$ is the photoelastic coefficient of the fiber (normally $P_{e}=0.22$), and $\lambda_{B0}$ is the unmodulated Bragg wavelength of FP sensor, so that the actual Bragg wavelength under modulation by ε(t) will become
(17)$$\lambda_{B}\left(t\right)=\lambda_{B0}+\Delta \lambda \left(t\right)$$

Reflection spectrum (transfer function) of a twin-FBG based FP sensor can be expressed as [43]
(18)$$\begin{array}{cc} R_{FP}\left(\lambda ,t\right) & =2{\left[\frac{\pi n_{1}L_{BG}}{\lambda }\sinh\frac{2n\left(\lambda -\lambda_{B}\left(t\right)\right)L_{BG}}{\lambda \lambda_{B}\left(t\right)}\right]}^{2}\times \left(1+\cos\frac{4\pi nL_{FP}}{\lambda }\right)\\ & =2R_{BG}\left(\lambda ,t\right)\times R_{m}\left(\lambda \right) \end{array}$$
where $n_{1}$ is the modulation amplitude of the effective refractive index of the grating, *n* is the mean refractive index of the fiber, $L_{BG}$ is the length of each grating. In Equation (Equation 18), $R_{FP}\left(\lambda ,t\right)$ contains two functions: $R_{BG}\left(\lambda ,t\right)$ and $R_{m}\left(\lambda \right)$. $R_{BG}\left(\lambda ,t\right)$ represents the envelope of reflection spectrum, while $R_{m}\left(\lambda \right)$, as a fringe function produced by the interference of reflected lights from two FBGs, represents an intensity modulation to $R_{BG}\left(\lambda ,t\right)$. $\lambda_{B}\left(t\right)$ determines the center position of $R_{FP}\left(\lambda ,t\right)$, while λ specifies the reflectance of FP sensor.

Following the changes of the fiber axial strain, the cavity length of FP sensor, $L_{FP}$, also is changed, resulting in a change of the fringe interval. However, since the relative change of grating pitch always matches up to that of the cavity length, due to both existing in same optical fiber, having $\Delta L_{FP}/L_{FP}=\Delta \Lambda /\Lambda$, the actual changes of the cavity length eventually do not bring substantive variations in $R_{m}\left(\lambda \right)$. Therefore, like traditional FBG sensors, the Bragg wavelength shift of FP sensor induced by ε(t) will only move $R_{FP}\left(\lambda ,t\right)$ in a linear way without obviously altering its envelope [43].

When a laser light with power $P_{0}$ and wavelength $\lambda_{0}$ close to $\lambda_{B0}$ illuminates the sensor, subjected to a power-frequency *E*-field $E_{0}\left(t\right)$, the returned signal light, expressed as $P\left(t\right)=P_{0}R_{FP}\left(\lambda_{0},t\right)$, will be a linear function of $E_{0}\left(t\right)$ through modulating $\lambda_{B}\left(t\right)$ by ε(t) following the AC *E*-field force F(t). As a result, the intensity of the interference signal detected by the photoelectric detection unit will be a sinusoidal signal with power frequency $f_{e}$, a phase lag of ϕ=π/2 to $E_{0}\left(t\right)$ and an amplitude proportional to the field strength $|E_{0}\left(t\right)|$. In this way, an all-fiber power-frequency *E*-field sensing is realized.

## 3. Sensor Fabrication and System Configuration

### 3.1. Sensor Fabrication

In our previous work, we totally made a dozen sensor prototypes with different working frequencies and packages used for testing sensor performances in various aspects. The twin-FBG sensors used in these prototypes were same in terms of their optical parameters: the reflectivity of 15%, grating length of about 1 mm, cavity length of 10 mm, and center wavelength of around 1542 nm. Figure 4a shows the spectrum of a twin-FBG sensor at T = 22.4 °C. The length of beam is variable, mainly depending on the working frequency, for example, it is ∼39 mm at 60 Hz or ∼45 mm at 50 Hz. The optimal length could be found through a vibration test at a required frequency by finely tailoring the beam length. Finally, the sensor was packaged into a glass tube for protecting it from external influences, e.g., the moisture, ionized air and impacts. It should be noted that as a kind of dielectric, the glass tube with a dielectric constant of 3∼10 is easy polarized when it is subjected to an *E*-field. The polarization charges distribute on both inner and outer walls of the glass tube with different densities, which may bring about some effects on the local *E*-field. For example, the field strength inside the glass tube is enhanced due to the larger curvature of inner wall on which the polarization charge density becomes higher. This feature had been confirmed in our previous experiments and was utilized in the sensor package design to improve the sensor sensitivity. Basic process flow for sensor fabrications is described as follows:Dry PI tubing in a chamber with humidity ≤20% RH and temperature at 110 °C for 4 h,Charge PI tubing by using a jig as shown in Figure 2a, imposing a 1000-V DC voltage on two electrodes and keeping this state at 80 °C for at least 1 h,Remove the DC voltage, keep PI tubing at 10 °C for 10 min and then take it out from the jig,Insert the twin-FBG fiber sensor into the charged PI tubing to constitute a composite cantilever beam (Figure 4b) with a pre-tailored length,Hold the beam with a plastic fastener (nozzle), fasten it by pushing a hoop toward the center of nozzle, and then fix them with the epoxy bond to form a sensor (see Figure 4c,d),Mount the sensor on a shaker oscillating in a frequency-scanning mode to check the resonant frequency of the sensor,Finely tailor the length of beam to maximize the vibrating amplitude at power frequency, andPackage the sensor into a glass tube containing desiccants and then seal this tube (see Figure 4e).

### 3.2. System Configuration

Figure 5 shows a schematic diagram of our detection system for interrogating the *E*-field sensor, in which a DFB (distributed feedback) laser diode is used as the light source of the system. The working wavelength of light source, always being locked at the center region of $R_{FP}\left(\lambda ,t\right)$, is dynamically controlled by a lock-in detection based feedback loop to real-time trace the slowly shifting of Bragg wavelength of the sensor, induced by ambient temperature fluctuating. For this, a small-amplitude, 1-MHz current signal $I_{s}\left(t\right)$ is superposed on the DC driving current of the laser diode. Here, it should be noted that a suitable electrical injection by $I_{s}\left(t\right)$ to the laser diode can effectively suppress the intensity noises arising in the interference signals due to the phase noises of the laser diode, so that the intensity of $I_{s}\left(t\right)$ injection should be adjustable. The sensor is connected to the system via a single-mode transmission optical fiber. The signal light P(t) returned from the sensor is detected by a photodetector and converted into the corresponding electrical signal which after passing through a low-pass filter (LPF) with a 200-Hz bandwidth and a voltage amplifier (Amp) is simultaneously sent to a digital oscilloscope for waveform observations as well as to a RMS (root means square) unit for signal amplitude detection. Owing to a very short length of FP cavity (in a dozen millimeters), the influences from the fluctuations of the states of polarization (SOP) of light beams propagating in the transmission fiber on the detection signals generally are negligible. Therefore, it becomes easier in actual applications to achieve a long-distance, remote *E*-field sensing.

## 4. Experimental Results

In this section, first of all, we will assess the dielectric charging effects of PI tubing samples, and then demonstrate the sensor performances in various aspects. Considering the data integrity, the data to be presented in following parts are from only a sensor prototype made with a charged PI tubing (code #068), which was employed in all experimental investigations, except for an experiment about a long-term running testing (durability), in which another prototype fabricated with a charged PI tubing (code #85) was selected.

### 4.1. Charging Effects of PI Tubing

Based on the proposed measuring procedure (see Figure 2b), an experiment for evaluating dielectric charging effects of PI tubing was carried out. An AC *E*-field meter, ME-3030B (GIGAHERTZ SOLUTIONS) with a detection frequency range of 16 Hz∼2 kHz, a maximum measurable field strength of 2 kV/m, and a resolution of 1 V/m, was used to measure the charge *E*-field generated by PI tubing with or without charges, vibrating at 21.3 Hz, excited by a shaker. Two 7-cm long PI tubing samples with different code numbers, sample #068 and sample #085, were used in this test. ME-3030B was placed below PI tubing with a 5-cm gap. The background field strength was measured, that was about 3 V/m. Whole devices were placed into an electrically grounded metal box in order to avoid EMIs from ambient environment. The measurement started at half an hour later after the end of a charging process (Step 1∼3 in process flow). The driving voltage of the shaker was changed from 0 to 2.0 V, which corresponded to a variation of vibrating amplitude from 0 to 8 mm. Figure 6a shows three measured charge *E*-field strengths with these two samples, in which it is clear that the charge *E*-field can arise only when PI tubing has been electrically charged, otherwise no field strength change can be observed. In both samples, the charge *E*-field strengths are proportional to the driving voltage $V_{s}$, that is the vibration amplitude, and linearly increases as $V_{s}\geq 0.8$ V. The slope (sensitivity) of field strength vs. $V_{s}$ in sample #085 is relatively large, which indicates that sample #085 had trapped much more charges than sample #068, due to the larger surface area sample #085 held, which helps to trap much more charges during the charging process.

Figure 6b shows another group of data about the charge *E*-field decaying. In this measurement, the strength of the charge *E*-field created by another PI tubing sample (code #068) was measured with ME-3030B immediately after the end of charging. From these data, it will be found that the field strength decreases rapidly in first hour. We can deduce a time constant (1/e time constant) from the data, which is about 1.2 h. After 3 h, the field strength finally stabilized around 100 V/m only with a relatively small variation within ±1 V/m. This phenomenon can be interpreted as follows: after finishing dielectric charging, in a long time, all forms of electric charges existing in PI tubing still are in a unstable transient state, accompanying with continuously discharging of surface charges and detrapping of space charge [27]. During this period, surface charges seem relatively easy to disappear due to the adsorption of surface water molecular, which causes the increase of electrical conductivity of the PI material [44]. So, only those space charges stored in the deeper laminar in PI tubing can remain finally. The decay time on discharging/detrapping of charges could be estimated with a formula in [27].

For an electrode-contact dielectric charging, it is possible that a polymer simultaneously traps the charges with opposite polarity on different surfaces. However, their densities and amounts may be very different. In experiments, we found that PI tubing after charging obviously trapped much more negative charges (electrons). Four photographs presented in Figure 7 show this observation result. In this experiment, a glass rod was positively charged by rubbing it with a silk sheet, and then placed near a PI tubing (code #085) with or without charges. Obviously, PI tubing with charges started to bend toward the glass rod by attraction forces induced by the charges staying on PI tubing and the glass rod, respectively, with opposite polarity.

### 4.2. Vibration Property of Sensor

Figure 8a,b are two frequency responses of a sensor prototype in the frontal (with a mark) and lateral directions, respectively, measured in 0∼100 Hz. This prototype was fabricated using a PI tubing (code #068). The test was carried out under a shaker excitation in a frequency-scanning mode with a detection system shown in Figure 5 to interrogate the sensor. The beam length of the sensor had been finely tailored to make the resonant peaks arise at 60 Hz in both directions, which can be identified in Figure 8a,b. The 3-dB width of resonant peak, Δf, in the frontal direction is 1 Hz and in the lateral direction 1.48 Hz, so that the corresponding *Q*-factors in both directions are Q=60 and Q=40.54, respectively, measured with [40]
(19)$$Q=\frac{f_{0}}{|f_{1}-f_{2}|}$$
where $f_{0}$ is the resonant frequency, $f_{1}$ and $f_{2}$ are frequencies at half power points on both sides of $f_{0}$, respectively. It is obvious that since Q≫1, when the sensor works in the resonant state, the enhancement of the sensitivity is considerably significant.

### 4.3. Detection Property of Sensor

An experimental setup to generate the stable AC electric field for senor performance testing is schematically shown in Figure 9a, which included two planar electrodes with the same area (45×35 cm${}^{2}$), separated by an air gap 2Δ, a HV transformer and a variable transformer (VAC). Planar electrodes were installed in parallel on the optics bench supported by two pieces of Bakelite (insulators). The sensor under test was placed between two electrodes with a distance *d* to an electrode. The output voltage of the HV transformer was directly applied to these two electrodes, while its input voltage was controlled by means of the VAC.

Figure 9b shows two signal waveforms. One is the *E*-field signal detected by our sensor; another is an AC ling signal picked up by the probe of the oscilloscope near an electrode. Figure 9c is a Lissajous figure formed by these two signals. Obviously, a phase difference between the *E*-field signal (representing the deflection of the cantilever beam) and the AC line signal (representing the *E*-field force) approximates $90^{o}$, indicating that this sensor indeed worked in a resonant mode.

Figure 10a is a set of the RMS output voltage (the signal RMS amplitude) against the imposed *E*-field strength $E_{0}$ or voltage $V_{\mathrm{HV}}$. In this result, obviously, the RMS output voltage is proportional to $E_{0}$ or $V_{\mathrm{HV}}$, linearly increasing in 0∼16.1 kV/m or in 0∼3.5 kV and saturating as $E_{0}\geq 32.5$ kV/m or $V_{\mathrm{HV}}\geq 7$ kV. Figure 10b is an *E*-field signal waveform and its FFT spectrum in 0∼200 Hz detected as $E_{0}=9.2$ kV/m. With $E_{0}$ increasing, as $E_{0}\geq 16.1$ kV/m, the *E*-field signal started to distort, resulting in the harmonic components to come up. Figure 11a,b are the other two *E*-field signal waveforms as well as their FFT spectra, obtained as $E_{0}=23$ kV/m and $E_{0}=36.8$ kV/m, respectively. Clearly, as $E_{0}$ increasing, the distortions in *E*-field signal waveforms become even more obvious and the high-order harmonic components (1st and 2nd peaks) shown in FFT spectra seem to get significant enhancements.

The sensitivity (detection output vs. imposed field strength) measured in 0∼16.1 kV/m (0∼3.5 kV) is $K_{s}=173.65$
μV/(V/m), which decreases with increasing of the *E*-field strength as $E_{0}\geq 16.1$ kV/m. In this work, we improved a setup previously used for investigating the signal-to-noise ratio (SNR) [24] and used it to investigate the ability of the sensor in detecting weak field strengths. The associated results are shown in Figure 12a,b.

An experimental setup is shown in the inset in Figure 12a, in which only a single electrode was used and imposed with a 1-Vpp AC voltage from an arbitrary waveform generator (AWG) to generate the weak *E*-field. The sensor was placed near the electrode with a 3-cm gap. Whole appliances were put into an electrically grounded metal box in order to block external *E*-field influences. The field strength measured with ME-3030B at sensor position under $V_{\mathrm{HV}}$ = 1 V${}_{\mathrm{pp}}$ was $E_{0}=12$ V/m. To avoid 60-Hz line signal interference, the frequency of AC voltage was set at 60.5 Hz in this measurement. The signal level at 60.5 Hz read from the FFT spectrum in Figure 12a is −20 dB. The FFT spectrum in a 100-Hz frequency span was computed by the digital oscilloscope with 2048 acquiring points and a bandwidth of about 0.1 Hz. The required acquisition time was about 10 s. The mean noise floor within 1-Hz band as a 0-V voltage was applied (Figure 12b) is about −54.4 dB, so that SNR = 34.4 dB at 60.5 Hz. Considering the resonant peak width of about 1 Hz, when the sensor works at 60 Hz (see Figure 8a), the actual SNR at 60 Hz will be 37.4 dB. Therefore, the minimum detectable field strength achieved with this sensor, calculated with [45]
(20)$$E_{min}=E_{0}10^{-\mathrm{SNR}/20}$$
is $E_{min}=0.162$ V/m.

### 4.4. Directionalities of Sensor

Angle dependencies of the sensitivity, that is, the directionalities of the sensor about its two principle axes, vertical and longitudinal axes, were investigated. For an *E*-field sensor, the directional characteristic in many applications often is required, where the field component in a particular direction needs to be measured. First, an angle dependency about sensor’s vertical axis or the *z*-axis (a coordinate adopted in Figure 9a) was investigated. The sensor was mounted horizontally on a turntable stage in front of an energized electrode and its azimuth angle θ was changed by rotating the stage from $-90^{o}$ to $+90^{o}$ as schematically shown in Figure 13a. The sensitivities at different azimuth directions were measured and are plotted in Figure 13a. From this result, clearly, this sensor is highly orientation dependent, and its normalized sensitivity varies from 0.0081 at $\theta =0^{o}$ to 0.9726 at $\theta =+90^{o}$ with a difference of about 21 dB.

Secondly, for investigating the angle dependency about the longitudinal axis of the sensor, the sensor was placed in parallel to an energized electrode and rotated counterclockwise about its longitudinal axis or the *x*-axis from $\varphi =0^{o}$ (the side with the minimum sensitivity) to $\varphi =90^{o}$ as schematically shown in Figure 13b. Meanwhile, assume that all field components are only in the horizontal direction, the sensitivities at different rotation angles were measured and are plotted in Figure 13b. From this result, clearly, the sensor also is rotation-angle dependent and its normalized sensitivity about the longitudinal axis varies from 0.114 at $\varphi =0^{o}$ to 1 at $\varphi =90^{o}$ with a difference of about 9.4 dB. This difference came mainly from the difference of vibration property of the sensor in different directions as well as from nonuniform distributions of space charge on PI tubing. It should be noted that this sensitivity-to-angle dependency can be controlled by adjusting the clamp pressure difference between two orthogonal sides of the cantilever beam using a specially designed nozzle.

### 4.5. System Stability Test

Stability of the detection system was investigated. Figure 14a shows two measured *E*-field signal waveforms with and without current signal $I_{s}\left(t\right)$ injection to the laser diode. In the upper waveform, without $I_{s}\left(t\right)$ injections, the signal amplitude fluctuating is obvious, however, in the lower waveform, which has been effectively suppressed through $I_{s}\left(t\right)$ injection. The injection intensity mainly depends on the type of laser diode to be used. Generally, the stronger the injection is, the more stable the system becomes. However, too strong injection will result in an obvious deviation of working wavelength as well as the decline of the sensitivity.

Figure 14b shows two signal traces recorded during 10 h for demonstrating the system performance in respect of automatically tracing the shifting of Bragg wavelength of the sensor when ambient temperature fluctuating obviously. In this experiment, the field strength consistently remained stable. The upper is a trace of error signals from the output of Lock-in amplifier, which reflects the variations of Bragg wavelength with temperature. The maximum variation magnitude of the error signal is about 0.3 V, which corresponds to a temperature change of about 10 °C. The lower trace represents the field strength signal recorded in the same period, in which, however, no significant fluctuation in the signal level can be observed, even the temperature changed very obviously.

### 4.6. Dynamic Responses of System

Dynamic responses of the detection system to external field strength variations in small and large scales were investigated, respectively. First, the sensor was placed in an *E*-field using a setup in Figure 9a with 2Δ = 40 cm and d=15 cm. The field strength around the senor was periodically changed by adjusting $V_{\mathrm{HV}}$ manually with the VAC from 0 V to 5 kV and then back to 0 V in a 500-V step with a rising/declining time of ≤5 s and a 50-s duration within a 1000-s measurement span. The measured result is illustrated in Figure 15a, in which it can be seen that the RMS output voltage well follows the field strength change in a linear manner.

Secondly, the sensor was placed in front of a single electrode with d=20 cm. The field strength was rapidly changed by applying a 9-kV pulse AC voltage with a 2.5-s duration to the electrode. Figure 15b is the corresponded pulse signal detected by the system, in which two time constants in the rising and declining parts of the pulse (between 10% and 90% of amplitude) can be deduced, which are 0.316 s and 1.273 s, respectively. The rising time is basically determined by the response time of RMS unit in the detection system, which usually is in the several hundreds of ms, and the declining time mainly depends on the hysteresis characteristic of the sensor. In principle, even if an external AC *E*-field force is removed, the sensor still is able to work for a while under the excitation of the inertia force, due to a very small damping coefficient held by the cantilever system. These investigated results reflect comprehensively the dynamic characteristics of the sensor and detection system to response external field strength variations in small or large scales. In addition, Figure 15c shows a detected *E*-field signal waveform under a 9-kV pulse AC voltage excitation, in which a transient response of about 1.2 s to the imposed pulse voltage can be identified.

### 4.7. Durability of Sensor

Durability of the sensor based on a long-term running test was investigated. A sensor prototype made with a PI tubing (code #085) and packaged with a glass tube was specified for this test. This sensor has a relatively low sensitivity, $K_{s}=123.32$
μV/(V/m), and a maximum unsaturated RMS output voltage up to 4.5 V. After being packaged, the sensor was immediately put into an electrically grounded metal box (see an inset in Figure 16a) for the test. The field strength and temperature inside the box remained stable. First, the field strength inside the box was monitored for 100 h by this specified sensor and is shown in Figure 16a. In this result, it can be observed that within the first 20 h, the detected signal level (RMS output voltage) continuously abates from its initial value (4.391 V) down close to a mean value (4.303 V) and afterwards stops decaying. It indicates that after the end of dielectric charging, in a long time, there still exists a discharging process in PI tubing. Similar process also can be found in Figure 6b, in which the charged PI tubing was tested under an open environment without any protective isolation, so that the discharging in that case was more fast due to the effects of the moisture in environment. Same procedure was conducted repeatedly for a year. Figure 16b is a collection of monthly averaged signal level recorded within a year. And what shown in an insert of Figure 16b is the detail of a 100-h signal trace recorded in the 8^th^ month. From these results, clearly, during a year, the signal level always oscillated about a mean voltage of 4.307 V with small fluctuations, diminishing with time, which indicates that space charge stored in PI tubing gradually became stable.

## 5. Applications of Sensor

Experiments using this sensor technology to actually measure the strength of *E*-field in various applications were conducted. In this section, preliminary experimental results will be demonstrated.

### 5.1. Measurements of Field Strength Distribution

Field strength distributions in free space around an electrode were investigated. In this experiment, a single electrode applied with a HV AC voltage was used to form an *E*-field distribution around it; the sensor fixed on a movable stage parallel to the electrode was horizontally moved; meanwhile the field strength at different distances was measured. Two sets of the recorded RMS output voltage against the distance *d* with different HV voltages ($V_{\mathrm{HV}}=1.5$ kV and $V_{\mathrm{HV}}=1$ kV) are illustrated in Figure 17a, from which, it may be observed that with *d* increasing, the RMS output voltages rapidly decrease in a manner of $d^{-2}$. The RMS output voltage against *d* at $V_{\mathrm{HV}}=1.5$ kV, however, is of little different to that at $V_{\mathrm{HV}}=1$ kV as d≤5 cm. This difference reflects the saturation property of the sensor when being subjected to a high-strength *E*-field. Same measurement with ME-3030B was carried out; the similar result can be identified in Figure 17b.

Distributions of the *E*-field in a space between two parallel planar-electrodes with different gaps were investigated based on a setup in Figure 9a. In this experiment, the electrode separation 2Δ was taken as a parameter, changing from 8 to 32 cm. The sensor was fixed on a movable stage horizontally moving between two planar electrodes applied with a 1-kV AC voltage. Meanwhile, the field strengths at different positions were measured, which then were normalized and are plotted in Figure 18a. In this result, it may be observed that the field strength in this separation space is not uniform and becomes the lowest at the center region, although in theory, if the electrode area is large enough, it can be expressed as $E_{0}=V_{\mathrm{HV}}/\left(2\Delta \right)$, uniformly distributing in this space. It was because that with the increase of a ratio of 2Δ to the electrode area, the bending of field lines toward to ground would become obvious, which, in turn, reduced the amount of the field component in the horizontal direction, which was detected by the sensor. As a comparison, similar measured results with ME-3030B can be identified in Figure 18b.

### 5.2. Application for Electric Discharge Sensing

Figure 19a is an *E*-field signal detected as the sensor was placed below an energized power cable in which continuous corona discharges occurred. Observing the signal waveform, it can be found that the distortions obviously appear on the topside of the waveform, which were caused by corona discharges which perturbed the surrounding *E*-field. It may be noted that since the occurrence of corona discharges always is phase-relevant to the AC line voltage which may result in a periodic ionization of air surrounding the cable, the field distortion induced by corona discharges seems to repeat following with the AC line voltage.

Figure 19b is another *E*-field signal waveform detected when the sensor was placed close to a power transformer in which electric sparks occasionally occurred. In this result, a short section of distorted waveform, marked with a red ellipse, can be identified, which was induced by electric sparks occurred inside the power transformer. It should be noted that the disturbing degree to the *E*-field and the duration of electric sparks are very different to those of corona discharges. Therefore, it becomes possible to distinguish these types of electric discharges occurring in the power equipment through real-time waveform analyses with *E*-field signals detected by our sensor.

### 5.3. Application for Human Presence Sensing

Human body as a sort of conductor can interfere with the *E*-field distributing in space. This feature may be utilized as a means of detecting the human presence (proximity sensing) when a person moves near an energized power equipment. Figure 20a presents a photograph showing an experimental arrangement for such kind of applications. In this experiment, a whiteboard imposed with a 1-kV AC voltage was utilized to simulate a power equipment to generate an *E*-field distribution around it. The sensor was placed in front of this whiteboard with a gap of 12 cm. Figure 20b shows a RMS output signal trace recorded during a 30-s period, in which two peaks with larger intensity fluctuating can be observed, which reflect the changes in the field strength, induced by a person when approaching the whiteboard. As a preset, an experimenter was required to approach the whiteboard twice with different paces and staying time in front of the whiteboard. In the signal trace, the first large peak represents such a scene in which the experimenter first walked toward the whiteboard at a fast pace (about 0.8 m/s), then stopped in front of it with a gap of about 20 cm from the experimenter’s chest and then turned back to leave quickly without staying. The second large peak represents another scene in which the experimenter first approached the whiteboard at a slower pace (about 0.6 m/s) and stood in front of it at 17 cm from the experimenter’s chest for about 3.5 s, and then left quickly. Multiple small peaks superposed on two large peaks can be identified, which reflects the changes in the effective area of the human body when the experimenter was walking, for example, swinging his/her arms and legs. These results well demonstrate a feasibility of this sensor technology applying in different measurement fields such as for human presence sensing.

## 6. Conclusions

In this article, we have presented a novel fiber-optic sensor for power-frequency *E*-field sensing. The basic concept, structure, fabrication process and operation principle of the sensor have been systematically introduced. The comprehensive experiments with two sensor prototypes have been carried out, which involved the assessment of dielectric charging effects of PI tubing, the characterization of the sensor and the investigation of detection system performances. A sensor prototype exhibited a high sensitivity of 173.65 μV/(V/m) with a minimal detectable field strength of 0.162 V/m, while another had a durability of continuous operation for over a year. Also we have demonstrated the feasibility of this sensor technology applying in different measurement fields with several actual application cases, such as for the measurements of field strength around the energized objects, the detection of electric discharges occurred in the power equipment, as well as for human presence sensing.

Major advantages of our sensor scheme are the simplicity of the actual sensor head in structure as well as in fabrication, and the intuition of detection signals, which can real-time reflect the tiny distortions of *E*-field in each cycle. However, an obvious disadvantage existing in the sensor, resulted from the cantilever beam structure, is its high sensitivity to external impacts and mechanical vibrations, especially to a continuous vibration at the power frequency. These influences may cause the sensor to produce the false detection outputs, if there is no suitable measure to be adopted to block them. How to solve this problem has become one of our tasks in the next stage. Further work will probably focus on further improvements of sensor performances in terms of the sensitivity, linearity, dynamic range and mechanical stability, as well as on the calibration of all sensor parameters.

## Figures and Tables

**Figure 1 sensors-19-01456-f001:**
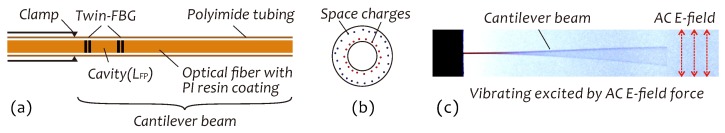
(**a**) A structure schematic of proposed sensor, (**b**) a sketch on space charge distributions inside PI tubing and (**c**) a photo of sensor vibrating in AC *E*-field.

**Figure 2 sensors-19-01456-f002:**
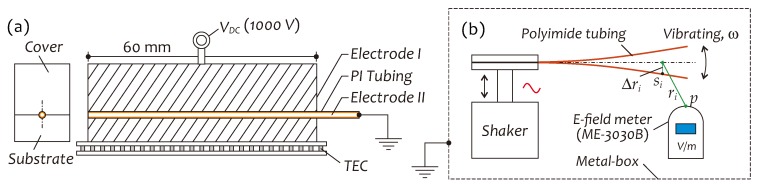
(**a**) A jig for PI tubing charging, (**b**) a setup for detecting charge field strength.

**Figure 3 sensors-19-01456-f003:**
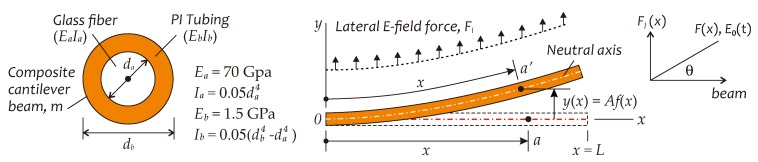
A structure schematic of composite cantilever beam adopted in *E*-field sensor.

**Figure 4 sensors-19-01456-f004:**
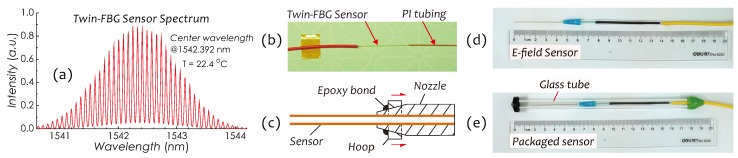
(**a**) Spectrum of a twin-FBG sensor; (**b**) a photo of twin-FBG sensor and PI tubing; (**c**) a schematic on parts assembly; (**d**) a photo of *E*-field sensor; and (**e**) a photo of packaged sensor.

**Figure 5 sensors-19-01456-f005:**
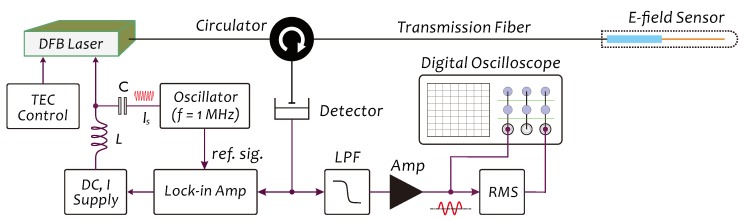
Schematic diagram of detection system configuration.

**Figure 6 sensors-19-01456-f006:**
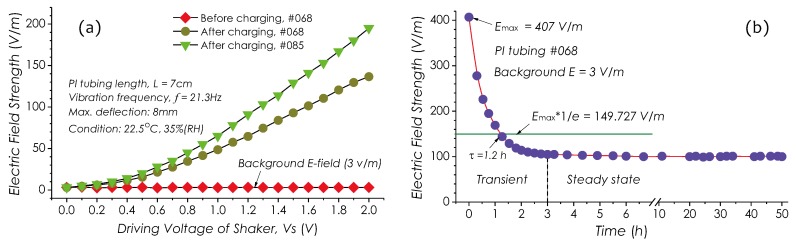
Results on charge *E*-field strengths measured with a proposed procedure. (**a**) Field strength vs. driving voltage of shaker and (**b**) charge *E*-field strength decaying with time.

**Figure 7 sensors-19-01456-f007:**
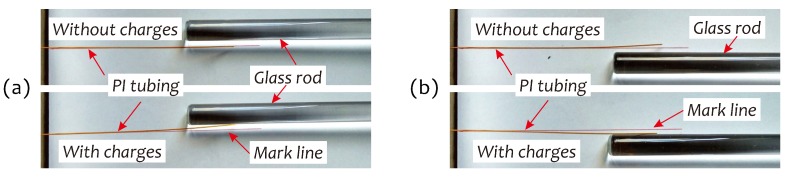
Four photographs for verifying charge polar in PI tubing. A positively charged glass rod was placed on left side (**a**) or right side (**b**) of PI tubing with or without charges. Mark line is to indicate original position of PI tubing.

**Figure 8 sensors-19-01456-f008:**
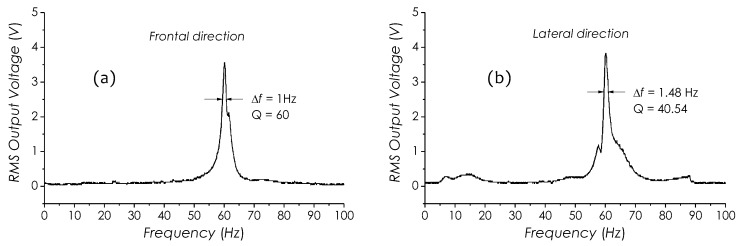
Frequency responses of sensor in 0∼100 Hz in mutually perpendicular directions, (**a**) frontal and (**b**) lateral.

**Figure 9 sensors-19-01456-f009:**
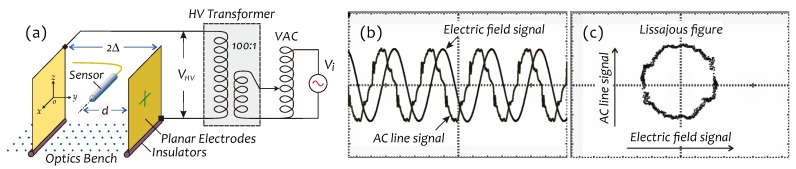
(**a**) Experimental setup for building an *E*-field environment. (**b**) *E*-field signal detected by our sensor and AC line signal detected by probe of oscilloscope, and (**c**) corresponding Lissajous figure.

**Figure 10 sensors-19-01456-f010:**
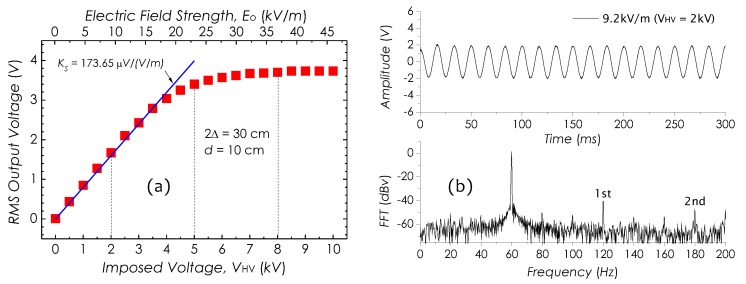
(**a**) RMS output voltage vs. imposed *E*-field strength $E_{0}$ or voltage $V_{\mathrm{HV}}$, and (**b**) a measured 60-Hz *E*-field signal waveform as $E_{0}=9.2$ kV/m ($V_{\mathrm{HV}}=2$ kV) and its FFT spectrum in 0∼200 Hz.

**Figure 11 sensors-19-01456-f011:**
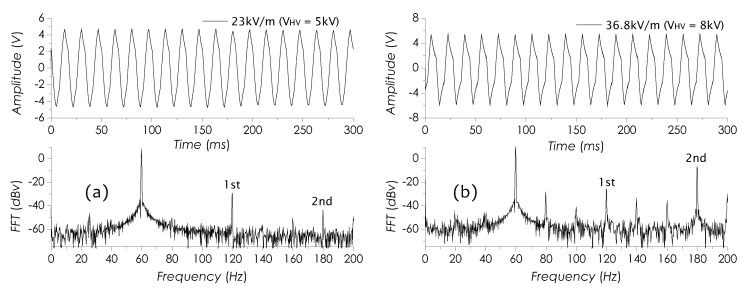
Measured *E*-field signals and FFT spectra at (**a**) $E_{0}=23$ kV/m and (**b**) $E_{0}=36.8$ kV/m.

**Figure 12 sensors-19-01456-f012:**
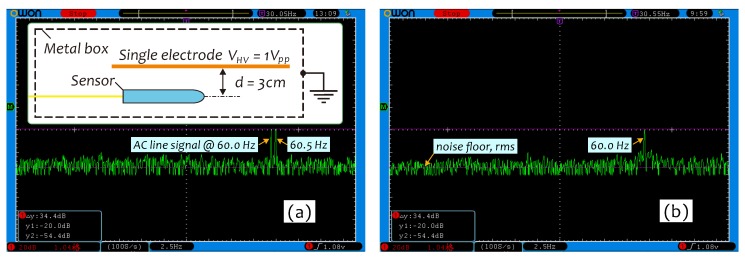
Two FFT spectra measured when (**a**) $V_{\mathrm{HV}}$ = 1 V${}_{\mathrm{pp}}$ and (**b**) $V_{\mathrm{HV}}=0$ V.

**Figure 13 sensors-19-01456-f013:**
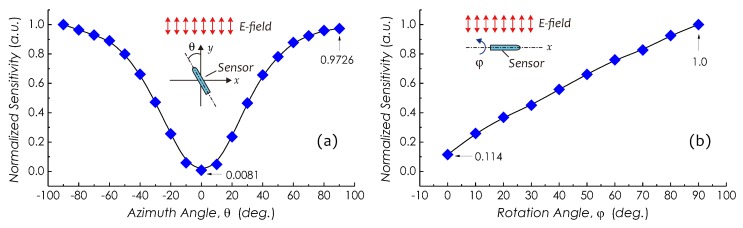
Directionalities of sensor about (**a**) vertical and (**b**) longitudinal axes.

**Figure 14 sensors-19-01456-f014:**
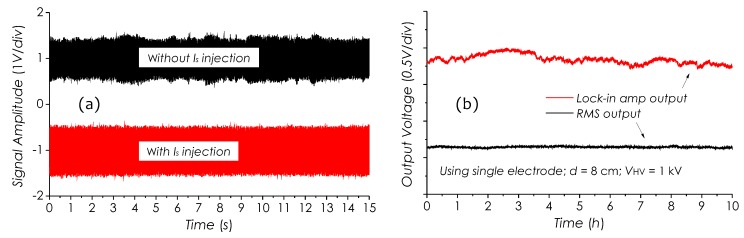
(**a**) Measured *E*-field signals with/without $I_{s}\left(t\right)$ injection, (**b**) error signal trace from lock-in amplifier and field strength trace recorded for 10 h.

**Figure 15 sensors-19-01456-f015:**
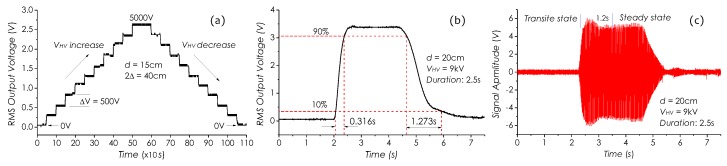
Dynamic responses of detection system, tested with (**a**) a 500-V step voltage, (**b**) a 9-kV pulse AC voltage. (**c**) is an *E*-field signal waveform under a 9-kV pulse AC voltage excitation.

**Figure 16 sensors-19-01456-f016:**
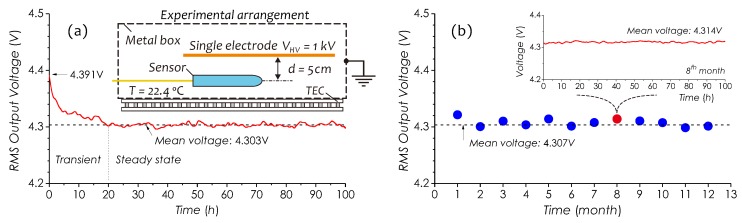
(**a**) Signal trace recorded in first 100 h, (**b**) a collection of all mean voltages of signal traces obtained in every month in a year and inset is a signal trace obtained in 8th month.

**Figure 17 sensors-19-01456-f017:**
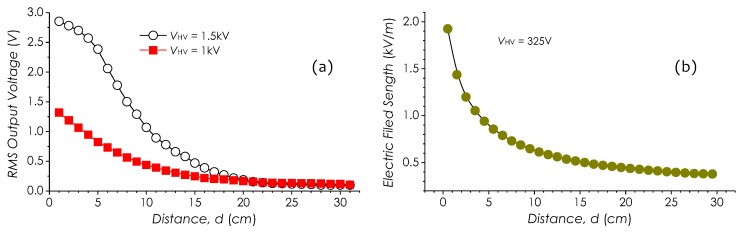
Measurements of field strength around an electrode, with (**a**) our sensor and (**b**) ME-3030B.

**Figure 18 sensors-19-01456-f018:**
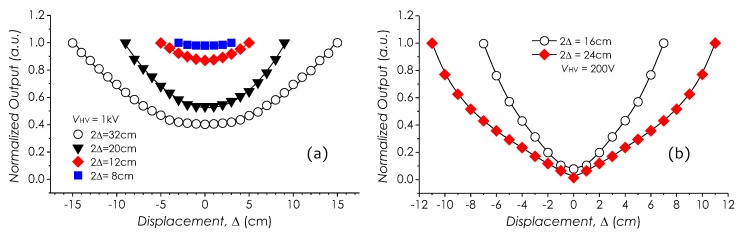
Field distributions between electrodes, measured with (**a**) our sensor and (**b**) ME-3030B.

**Figure 19 sensors-19-01456-f019:**
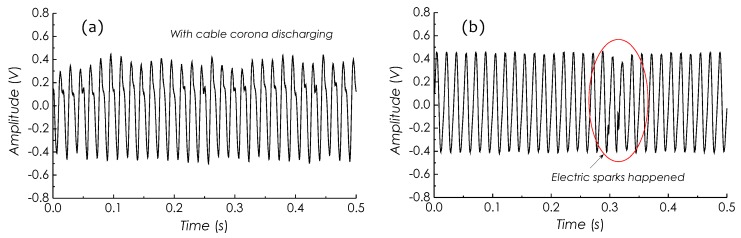
Measured *E*-field signal waveforms, (**a**) with corona discharges and (**b**) with electric sparks.

**Figure 20 sensors-19-01456-f020:**
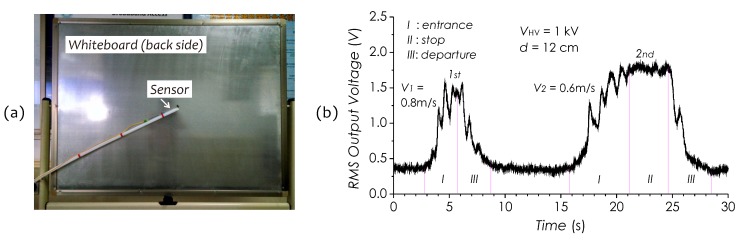
(**a**) A photograph of experimental arrangement used for human presence sensing. (**b**) RMS output signal trace recorded during a 30-s period.

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
