# Peer review of "Power-Frequency Electric Field Sensing Utilizing a Twin-FBG Fabry–Perot Interferometer and Polyimide Tubing with Space Charge as Field Sensing Element [Author-notes fn1-sensors-19-01456]"

_sensors, 2019, doi:10.3390/s19061456_

Reviewer 1 Report

This is an interesting paper with however many details are inconsistent.

The English needs considerable improvement, grammar is often wrong, sentences poorly constructed, and very often the wrong part of speech is used:  The whole manuscript requires expert editing to reach acceptable standards, with far more corrections than can be expected from a scientific reviewer.

Scientific issues

In the abstract and conclusions, you say sensitivity is 173.65 V / V/m with a minimal detectable field strength of 0.162 V/m

At thin minimum electric field strength, the measurement would be 176*0.16 = 28 V.

However, all the data plotted has output in the single digit Volt level (Figures 8,10, 11, 14,15,16,17,19,20).  Figure 10 appears to me to have a slope about 1 V / 5 kV/m = 0.0002 V / V/m

The stated sensitivity is incompatible with the data presented.  This needs to be fixed.

P7 what is “vibrar”?

L170 to 171:  I do not follow this logic.  If the field outside has value 1, then the field in the glass is increased to 3, which increase is reversed for the electric field inside the glass tube.  I cannot agree that the glass is “very helpful for improving the sensitivity”

L245: do you mean electrons and not electronics?  In this region, the word “charged” is used when the correct grammar would be one of “charge” or “charging”.  Figure 7 caption and the paragraph below have appalling English.

P12   The description in the paragraph below figure 10 (and captions for Figure 11) and much of the text following needs to be in terms of Electric field, not VHV since the sensor is an Electric field sensor, and the value of the source voltage irrelevant… changing distances will change the electric field even if VHV remains the same.  VHV controls the Electric field, but is NOT a parameter that should be used as the independent variable in the discussion.  These should ALL be in E (V/m).

The comparison in Figure 18 is worse than it looks, because the vertical axes are different.  These two plots (a) and (b) need to be on the same scale.

L415 Should this be 0.6 m/s and not 6 m/s?  Was all of the experimenter at 17 cm, or just his chest? Or what?

L433:  I do not understand this sentence.  What are “detail sections”?  What is the “distortion and abrupt change of the field strength”?

Author Response

I very appreciated your comments, thank you so much!

My response is as follows:  

1/ 

The English needs considerable improvement, grammar is often wrong, sentences poorly constructed, and very often the wrong part of speech is used:  The whole manuscript requires expert editing to reach acceptable standards, with far more corrections than can be expected from a scientific reviewer.

ans:

Yes, I will send this manuscript later to MDPI to receive an enhanced grammar checks and sentence corrections.

2/

In the abstract and conclusions, you say sensitivity is 173.65 V / V/m with a minimal detectable field strength of 0.162 V/m

ans: 

I’ve missed a unit μV in this sensitivity, and the correct one should be 173.65μV / V/m that is the slope of the fitting line in Figure 10. In the case for weak electric field measurements, the detection signal voltage for the minimum field strength of 0.162 V/m is 173.65*0.162μV = 28.13μV.

3/

At thin minimum electric field strength, the measurement would be 176*0.16 = 28 V.

However, all the data plotted has output in the single digit Volt level (Figures 8,10, 11, 14,15,16,17,19,20).  Figure 10 appears to me to have a slope about 1 V / 5 kV/m = 0.0002 V / V/m

ans:

The slope of fitting line in Figure 10 is 173.65μV / V/m.

The sensor output is a measured value on the electric field strength through the detection of AC field signal amplitude (RMS detection), so that it is often expressed in the single digit volt lever.

After taking a fine calibration for all sensing parameters with the standard field meter following a strict measurement process, the output of the sensor finally can be expressed in field strength (V/m) unit, like that adopted in commercial field meters.

4/ 

P7 what is “vibrar”?

ans:

“vibrar” is a unit adopted by Dr. C.F. Beards in his book “In Structural Vibration: Analysis and Damping” to express a dimensionless vibration intensity. I attached two pages from this book to explain this. However, this unit seems to be not necessary, so that in equation 13, I simply remove it for simplicity.

5/

L170 to 171:  I do not follow this logic.  If the field outside has value 1, then the field in the glass is increased to 3, which increase is reversed for the electric field inside the glass tube.  I cannot agree that the glass is “very helpful for improving the sensitivity”.

ans:

As one property of dielectric material (glass tube), the electric field strength inside the dielectric always is zero when the dielectric material is subjected to an external electric field. It may result in the variations of the field line density (field distortions) with irregular distributions of field strength on different surfaces of dielectric material with the irregular shape, for example, with different curvatures on different sides. Therefore, the field strength inside the glass tube with the larger curvature will be enhanced. I attached a sketch for explaining this phenomenon. Considering that it may cause the difficulty in understanding by other readers or may have to use much more sentences to explain this concept. For simplicity, I finally decided to remove this expression from the manuscript. This also is one of our works in next step and will be investigated extensively.

6/ 

L245: do you mean electrons and not electronics?  In this region, the word “charged” is used when the correct grammar would be one of “charge” or “charging”.  Figure 7 caption and the paragraph below have appalling English. 

ans:

This is my fault, here, the correct word should be “electrons” and not “electronics”. Also, I’ve corrected another misused word “charged”. It should be “charging”.

7/

P12   The description in the paragraph below figure 10 (and captions for Figure 11) and much of the text following needs to be in terms of Electric field, not VHV since the sensor is an Electric field sensor, and the value of the source voltage irrelevant… changing distances will change the electric field even if VHV remains the same.  VHV controls the Electric field, but is NOT a parameter that should be used as the independent variable in the discussion.  These should ALL be in E (V/m).

ans:

I’ve replaced the variable “VHV” in Figure 10 and Figure 11 with electric field strength (V/m).

8/

The comparison in Figure 18 is worse than it looks, because the vertical axes are different.  These two plots (a) and (b) need to be on the same scale.

ans:

I’ve adjected the scale in plots (a) to make two plots in the same scale.

9/

L415 Should this be 0.6 m/s and not 6 m/s?  Was all of the experimenter at 17 cm, or just his chest? Or what?

ans:

It should be 0.6m/s and not 6 m/s. I’ve corrected this mistake. The 17 cm should be a gap between the whiteboard and experimenter’s chest. I’ve added this information in my manuscript.

10/

L433:  I do not understand this sentence.  What are “detail sections”?  What is the “distortion and abrupt change of the field strength”?

ans:

I’ve replaced this sentence “detail sections” with anther expression “in each cycle”.

11/

What is the “distortion and abrupt change of the field strength”?

ans:

I’ve removed this inappropriate expression from the manuscript.

Manuscript will be received further grammar checks by English experts in MDPI later.

Reviewer 2 Report

This manuscript describes and all-optical electric filed sensors that is based on measurement of electric force assert to a longer section of fiber coated by PI. PI is charged to provide electric field response. Authors provided detail sensor description and analysis and also good experimental characterization/support. The approach is novel and interesting; thus I recommend publication after revision. However, I think paper needs very careful revisions as there are several technical inconsistences as well as language flaws (grammatical,  style, and use of words, i.e. “vibrar”, “detailed sections”, there are many similar occasions). Authors should also re-check reported experimental values for consistencies. For example, author claim sensitivity of 0.162 V/m. Authors however do not report at what measurement bandwidth (bandwidth and resolution are usually inversely proportional). They use FFT to extract component of the signal that corresponds to the electric field excitation freqency, however they do not report on the length and time of acquisition of the signal. The longer they acquire the signal before performing FFT, the higher the resolution. Based on very narrow spectrum within in Fig 12, it seems the acquisition time was very long, thus I am not sure reported resolution of 0.162 V/m is a realistic value (i.e. it is mostly possible to show very high resolution if acquisition time would go into minute or even range beyond that). I think authors should report resolution when acquisition time which does not exceed 1 s.

Author Response

I very appreciated your helpful comments!

Your questions are:

I think paper needs very careful revisions as there are several technical inconsistences as well as language flaws (grammatical,  style, and use of words, i.e. “vibrar”, “detailed sections”, there are many similar occasions). Authors should also re-check reported experimental values for consistencies. For example, author claim sensitivity of 0.162 V/m. Authors however do not report at what measurement bandwidth (bandwidth and resolution are usually inversely proportional). They use FFT to extract component of the signal that corresponds to the electric field excitation frequency, however they do not report on the length and time of acquisition of the signal. The longer they acquire the signal before performing FFT, the higher the resolution. Based on very narrow spectrum within in Fig 12, it seems the acquisition time was very long, thus I am not sure reported resolution of 0.162 V/m is a realistic value (i.e. it is mostly possible to show very high resolution if acquisition time would go into minute or even range beyond that). I think authors should report resolution when acquisition time which does not exceed 1 s.

My response is as follows:

1/

After re-checking all experimental data, I’ve made some corrections. The manuscript will receive a strict check in the uses of English words and sentences by a language expert in MDPI later.

2/

“vibrar” is a unit adopted by Dr. C.F. Beards in his book “In Structural Vibration: Analysis and Damping” to express a dimensionless vibration intensity. I attached two pages from this book to explain this. However, this unit is not necessary to be used, so I simply remove it from my manuscript.

3/

I’ve replaced the sentence “detail sections” with anther sentence “in each cycle”.

4/

I’ve added the relevant information in my manuscript on the acquisition time, bandwidth and data numbers for performing an FFT operation. FFT spectrum shown in Fig. 12 was obtained in the oscilloscope with a 10-s acquisition time, a 100-Hz frequency span, 2048 acquiring points and an about 0.1-Hz bandwidth.

When the acquisition time is reduced to 1 s, the bandwidth will increase to 1 Hz. However, under this condition, two frequency components at 60 Hz (background electric field) and at 60.5 Hz (an electric field to be measured) will overlapped. Therefore, the measured result will inevitably be influenced by the unwanted background electric field which is hard to remove and varies randomly. In our previous experiment, we ever obtained a minimum detectable field strength of about 3.3 V/m within 4 s and with a bandwidth of 0.25 Hz.
